# Nocturnal hypercapnia with daytime normocapnia in patients with advanced pulmonary arterial hypertension awaiting lung transplantation

**Yoshinari Nakatsuka[1], Toyofumi Chen-Yoshikawa[2], Hideyuki Kinoshita[3], Akihiro Aoyama[4], Hiroyasu Kubo[5], Kimihiko Murase[1], Satoshi Hamada[6], Hirofumi Takeyama[1], Takuma Minami[7], Naomi Takahashi[1], Kiminobu Tanizawa[7], Tomohiro Handa[6], Toyohiro Hirai[7], Hiroshi Date[2], Kazuo Chin** [1] *

1 Department of Respiratory Care and Sleep Medicine, Graduate School of Medicine, Kyoto University, Kyoto, Japan, 2 Department of Thoracic Surgery, Kyoto University Hospital, Kyoto, Japan, 3 Department of Cardiovascular Medicine, Graduate School of Medicine, Kyoto University, Kyoto, Japan, 4 Department of Thoracic Surgery, Kobe City Medical Center General Hospital, Kobe, Japan, 5 Division of Medical Equipment, Kyoto University Hospital, Kyoto, Japan, 6 Department of Advanced Medicine for Respiratory Failure, Graduate School of Medicine, Kyoto University, Kyoto, Japan, 7 Department of Respiratory Medicine, Graduate School of Medicine, Kyoto University, Kyoto, Japan

* chink@kuhp.kyoto-u.ac.jp

**Data Availability Statement:** All relevant data are within the manuscript and its Supporting Information files.

## Abstract

### Background

Pulmonary arterial hypertension (PAH) is frequently complicated by sleep disordered breathing (SDB), and previous studies have largely focused on hypoxemic SDB. Even though nocturnal hypercapnia was shown to exacerbate pulmonary hypertension, the clinical significance of nocturnal hypercapnia among PAH patients has been scarcely investigated.

### Method

Seventeen patients with PAH were identified from 246 consecutive patients referred to Kyoto University Hospital for the evaluation of lung transplant registration from January 2010 to December 2017. Included in this study were 13 patients whose nocturnal transcutaneous carbon dioxide partial pressure ($PtcCO_2$) monitoring data were available. Nocturnal hypercapnia was diagnosed according to the guidelines of the American Academy of Sleep Medicine. Associations of nocturnal $PtcCO_2$ measurements with clinical features, the findings of right heart catheterization and pulmonary function parameters were evaluated.

### Results

Nocturnal hypercapnia was diagnosed in six patients (46.2%), while no patient had daytime hypercapnia. Of note, nocturnal hypercapnia was found for 5 out of 6 patients with idiopathic PAH (83.3%). Mean nocturnal $PtcCO_2$ levels correlated negatively with the percentage of predicted total lung capacity (TLC), and positively with cardiac output and cardiac index.

**Funding:** This work was supported in part by grants from the Japanese Ministry of Education, Culture, Sports, Science and Technology (26293198, 17H04182), the Intractable Respiratory Diseases and Pulmonary Hypertension Research Group, the Ministry of Health, Labor and Welfare in Japan (H29-intractable diseases-general-027), the Research Foundation for Healthy Aging, Health, Labour and Welfare Sciences Research Grants, Research on Region Medical from the Ministry of Health, Labor and Welfare of Japan (H28-iryo-ippan-016, H30-iryo-ippan-009). The funders had no role in study design, data collection and analysis, decision to publish, or preparation of the manuscript.

**Competing interests:** I have read the journal's policy and the authors of this manuscript have the following competing interests: Yoshinari Nakatsuka and Naomi Takahashi reports grants from Philips-Respironics, grants from ResMed Japan, grants from Fukuda Denshi, grants from Fukuda Lifetec Keiji. Kimihiko Murase and Hirofumi Takeyama reports grants from Philips-Respironics, grants from Teijin Pharma, grants from Fukuda Denshi, grants from Fukuda Lifetec Keiji. Kazuo Chin reports grants and personal fees from Philips-Respironics, grants and personal fees from Teijin Pharma, grants and personal fees from Fukuda Denshi, grants and personal fees from Fukuda Lifetec Keiji, grants from KYORIN Pharmaceutical Co., Ltd, grants from Nippon Boehringer Ingelheim Co., Ltd, grants and personal fees from GlaxoSmithKline, personal fees from MSD, personal fees from ResMed, personal fees from Astellas Pharma, personal fees from Eisai Co., Ltd. Hideyuki Kinoshita reports personal fees from Actelion Pharmaceuticals Japan Ltd., Nippon Shinyaku Co., Ltd, Bayer Yakuhin, Ltd., and research grant from Bayer Yakuhin, Ltd. Toyofumi Chen-Yoshikawa, Akihiro Aoyama, Hiroyasu Kubo, Satoshi Hamada, Takuma Minami, Kiminobu Tanizawa, Tomohiro Handa, Toyohiro Hirai and Hiroshi Date declare no potential conflict of interests. The Department of Respiratory Care and Sleep Control Medicine is funded by endowments from Philips-Respironics, ResMed, Fukuda Denshi and Fukuda Lifetec-Keiji to Kyoto University. These competing interests do not alter our adherence to PLOS ONE policies on sharing data and materials.

## Conclusion

Nocturnal hypercapnia was prevalent among advanced PAH patients who were waiting for lung transplantation, and associated with %TLC. Nocturnal hypercapnia was associated with the increase in cardiac output, which might potentially worsen pulmonary hypertension especially during sleep. Further studies are needed to investigate hemodynamics during sleep and to clarify whether nocturnal hypercapnia can be a therapeutic target for PAH patients.

## Introduction

Pulmonary hypertension (PH) is a syndrome resulting from multiple clinical conditions and disorders. Pulmonary arterial hypertension (PAH), corresponding to Group 1 of the Nice classification, is a group of diseases directly affecting pulmonary arterial resistance. PAH includes idiopathic PAH (IPAH), congenital heart disease-associated PH and connective tissue disease-associated PH [1, 2].

PAH is the fourth leading cause of the necessity for lung transplantation worldwide [3]. Although the recent development of medications to lower pulmonary arterial pressure (PAP) has substantially improved the prognosis of PAH [4], about 10% of PAH patients were reported to die within one year after the diagnosis [5]. The prognosis is especially poor for those with persistently high PAP despite extensive therapy [6], and considered for the candidates of lung transplantation [2]. While lung transplantation has been established as a therapeutic option for advanced PAH, interventions that can improve the hemodynamics are also required for those on the waiting-list for lung transplantation.

Disturbance in gas exchange is one of the major disorders in PH patients [2]. Previous studies showed that hypoxia and hypercapnia worsen PAH [7]. Indeed, hypoxia causes constriction of the pulmonary artery and increases PAP [8]. Similarly, hypercapnia also contributes to the elevation of PAP, and it has been proposed that increases in pulmonary vascular resistance (PVR) and cardiac output (CO) additively work as the pathogenesis of PAH(11–13). In addition, respiratory acidosis due to high $CO_2$ partial pressure can also cause PAP escalation [9, 10].

While the disturbance in pulmonary function causes alveolar hypoventilation and chronic $CO_2$ retention, sleep disordered breathing (SDB), including sleep apnea syndrome (SAS), may result in nocturnal hypoventilation and hypercapnia [11]. Previous studies suggested that SAS as well as nocturnal hypercapnia were associated with the escalation of PAP [12–14] and that positive pressure ventilation during sleep could lower daytime PAP [15–17]. In addition, SAS is a common morbidity for patients with PAH [18], thus nocturnal hypercapnia due to SDB may play a significant role in the pathogenesis of PAH. However, thus far no study has investigated the associations between nocturnal hypercapnia and clinical indices for refractory PAH patients so far.

The objective of this study is to evaluate the clinical relevance of nocturnal hypercapnia in advanced PAH being considered for lung transplantation. We hypothesized that nocturnal hypercapnia would be associated with worsened PH in these most severely affected patients. We evaluated nocturnal hypercapnia by using a transcutaneous carbon dioxide partial pressure (PtcCO$_2$) monitoring system, and investigated the associations of nocturnal hypercapnia with clinical and hemodynamic parameters.

## Materials and methods

### Study patients

We reviewed 246 consecutive cases that were referred to Kyoto University Hospital for the evaluation of suitability for lung transplantation from January 2010 to December 2017. We identified 17 patients who were classified as having Group 1 PH according to the current guidelines [2]. The diagnosis of PAH was made at the primary hospitals, and confirmed at Kyoto University. We then identified 13 patients whose pre-operative whole-night $PtcCO_2$ monitoring data were available and performed further analyses. We included patients irrespective of whether they were ultimately enrolled as candidates for lung transplantation. For all included cases right heart catheterization was performed, and those with post-capillary PH or thromboembolic pulmonary hypertension was excluded. High-resolution computed tomography was performed in all cases and patients with any comorbid lung diseases that might result in PH were denied as study patients. Written informed consent was obtained from most of the study participants. For the patients whose contact information was lost, we announced the conduct of this study on our institutional website and asked to contact us if they disagreed with our data access. The Kyoto University Hospital Institutional Review Board approved this study including ethical policy for data access (R1287).

### Continuous evaluation of $PtcCO_2$

According the manual of the American Academy of Sleep Medicine (AASM), we used $PtcCO_2$ as a surrogate measurement of $PaCO_2$ [19]. We utilized the TOSCA device (TOSCA measurement system and TOSCA 500 monitor, Linde Medical Sensors, Basel, Switzerland) because it has been demonstrated to have good accuracy in adult patients compared with blood gas analysis [20–22]. $SpO_2$ was simultaneously monitored. Nocturnal hypercapnia was diagnosed according to the guidelines of AASM; i.e. $PtcCO_2$ >55 mmHg for ≥10 minutes or an increase in $PtcCO_2$ ≥10 mmHg in comparison to an awake supine value exceeding 50 mmHg for ≥10 minutes [23]. Daytime hypercapnia was diagnosed in cases in which $PaCO_2$ evaluated by daytime arterial blood gas (ABG) sampling at rest was higher than 45 mmHg. ABG data from samples obtained at the nearest day to the $PtcCO_2$ monitoring were utilized for the analysis.

### Data collection and analysis

The clinical data were retrospectively collected from medical records. Patients were grouped according to the presence or absence of nocturnal hypercapnia. Analyzed parameters included results of right heart catheterization, pulmonary function tests and laboratory tests. The correlations between right heart catheter parameters and mean nocturnal $PctCO_2$ were also evaluated. Data from the nearest day to the $PtcCO_2$ monitoring were utilized for the analyses.

### Statistical analyses

Data are summarized as median (range) and number (percentage). We considered that the number of study patients was too small to determine whether each parameter followed normal distribution or not, therefore we employed non-parametric statistics for analyses. Fisher's exact test, Mann-Whitney's U test and Spearman's rank correlation test were used as appropriate. EZR (Saitama Medical Centre, Jichi Medical University, Saitama, Japan) [24], which is a graphical user interface for R (The R Foundation for Statistical Computing, Vienna, Austria), or GraphPad Prism version 7.00 for Windows (GraphPad Software, La Jolla, CA, USA, www.graphpad.com) was used for statistical analyses. A *P*-value less than 0.05 was determined to be statistically significant.

## Results

### Characteristics of patients with and without nocturnal hypercapnia

Thirteen PAH patients with available overnight PtcCO2 measurements were identified for this study. Among them, eight patients were registered for cadaveric lung transplantation, three patients received living-donor lung transplantation, one patient died shortly after the evaluation before the registration and the other patient was not registered due to an improved response to medical treatment. As shown in Table 1, six out of 13 patients (46.2%) had nocturnal hypercapnia (representative test results are presented in S1 Fig). There were no differences between patients with and without nocturnal hypercapnia in terms of age, sex, body mass index or blood BNP level. The treatment procedure was also similar, although the introduction of intravenous epoprostenol treatment was relatively frequent in patients with nocturnal hypercapnia (86.6% vs. 28.6%, P = 0.103). Two patients had received prior chest surgery, both of which were repair surgery for congenital heart diseases, neither of these patients exhibited nocturnal hypercapnia.

All patients were receiving long-term oxygen therapy. Differences in the rates of death and/ or transplantation were not significant.

Notably, of the 6 patients with nocturnal hypercapnia, 5 had IPAH (83.3%). A comparison of the mean and maximum $PtcCO_2$ levels at night between IPAH patients and non-IPAH patients showed that mean and maximum $PtcCO_2$ at night were significantly higher among the IPAH patients (Fig 1).

### Parameters of respiratory status and pulmonary function in patients with and without nocturnal hypercapnia

We next investigated the characteristics of the respiratory status and pulmonary function of patients with and without nocturnal hypercapnia (Table 2). Seven of the 13 patients had a decrease in %FVC <80%, while there were no significant differences in %FVC, %FEV1 and % DLco between the two groups. On the other hand, we found that the %TLC and %PEF in the nocturnal hypercapnia group was significantly lower than in the group without nocturnal hypercapnia; also patients with nocturnal hypercapnia had a significantly higher mean and maximum $PtcCO_2$. All the patients were prescribed long-term oxygen therapy including usage during sleep. However, an increase in 3%ODI, which is generally regarded as a representative indicator of SDB, was rarely observed in either group. Daytime $PaCO_2$ was equivalent in both groups, and daytime hypercapnia was not observed in any patient in the present study.

We further investigated whether mean $PtcCO_2$ was associated with pulmonary function indexes. We found negative correlations between mean $PtcCO_2$ at night and %TLC (Fig 2).

### Correlations between nocturnal hypercapnia and hemodynamic parameters

Among the parameters of hemodynamic status, we found that the patients with nocturnal hypercapnia had a significantly higher levels of cardiac output (CO) and cardiac index (CI) measured by right heart catheterization (P = 0.012 and P = 0.015, respectively). However, there were no significant differences between groups in other right heart catheter parameters such as mean, systolic and diastolic PAP, mean PCWP and PVR (Table 3). Investigation of the correlations between the mean $PtcCO_2$ at night and the CO and CI showed strong correlations (Fig 3A and 3B).

**Table 1. Background data on study patients.**

| | Nocturnal hypercapnia (-) | Nocturnal hypercapnia (+) | *P*-value |
|---|---|---|---|
| Number | 7 | 6 | |
| Age | 31.00 [12.00, 59.00] | 26.00 [14.00, 53.00] | 0.568 |
| Male sex (%) | 3 [42.9] | 0 [0.0] | 0.192 |
| Diagnosis | IPAH:1 | Eisenmenger synd: 1 | |
| | Systemic sclerosis: 1 | | |
| | Takayasu Disease: 1 | | |
| | PVOD: 1 | IPAH:5 | |
| | Eisenmeger synd: 3 | | |
| Previous thoracic surgery | 2 | 0 | 0.462 |
| Nasal 0xgen supplementation (L/min) | 2.00 [1.00, 7.00] | 2.00 [1.00, 4.00] | 0.348 |
| Body mass index | 18.37 [15.35, 25.57] | 18.19 [13.29, 30.32] | 0.886 |
| Treatment | | | |
| Number of oral anti-PH medication | 2.00 [1.00, 4.00] | 2.00 [1.00, 3.00] | 0.879 |
| Diuretics usage | 6 [85.7] | 5 [83.3] | >0.99 |
| IV Epoprostenol usage | 2 [28.6] | 5 [83.3] | 0.103 |
| IV Epoprostenol dose (ng/kg/min) | 35.5 [0.9, 70] | 51.0 [45, 200.0] | 0.571 |
| Laboratory tests | | | |
| BNP (pg/ml) | 98.70 [11.90, 586.90] | 127.90 [10.10, 170.50] | 0.465 |
| Hemoglobin (g/dl) | 12.00 [10.00, 24.70] | 11.30 [9.30, 19.90] | 0.431 |
| Lung transplantation | 2 [28.6] | 3 [50.0] | 0.592 |
| Death | 2 [28.6] | 0 [0.0] | 0.462 |

Data are presented as number [%] or median [range]. Mann-Whitney's U-test or Fisher's exact test was used for statistical analyses. IPAH: idiopathic pulmonary arterial hypertension, PVOD: pulmonary venous occlusive disease.

## Discussion

In the present study nocturnal hypercapnia was observed in nearly half of the PAH patients, as well as in most of those with IPAH, who were considered for lung transplantation. Mean $PtcCO_2$ at night correlated negatively with %TLC. Higher $PtcCO_2$ was associated with increased CO and CI. This is the first study to investigate the clinical significance of nocturnal hypercapnia in PAH patients.

Nocturnal hypercapnia results mainly from alveolar hypoventilation during sleep. The pathogenesis of nocturnal hypercapnia is comprised of the impairment of pulmonary function and an insufficient response against hypoxia/hypercapnia [11]. Although obesity hypoventilation syndrome is a major cause of nocturnal hypercapnia [11], the BMI in the majority of the current study patients was lower than normal (18.5 kg/m$^2$). Previously it was revealed that a low BMI affects the respiratory muscle mass, which results in the impairment of muscle performance [25] and a proportional decline in respiratory pressure [26]. In the present study, low values for %FVC, %TLC and %PEF were the most distinct features of nocturnal hypercapnia. Total lung capacity is determined by lung elastic recoil, chest wall elastic recoil and inspiratory muscle power, whereas expiratory muscle power and airway obstruction largely affects PEF [27]. Given that in this study restrictive and obstructive pulmonary diseases were denied by computed tomography and pulmonary function test, and lung and chest wall elastic recoils are not changed in patients with PAH in general, the decrease in %TLC and %PEF would be due to a decline in inspiratory and expiratory muscle power, respectively [27]. Other potential factors for the restriction of chest movement, such as previous thoracic surgery or skin stiffness

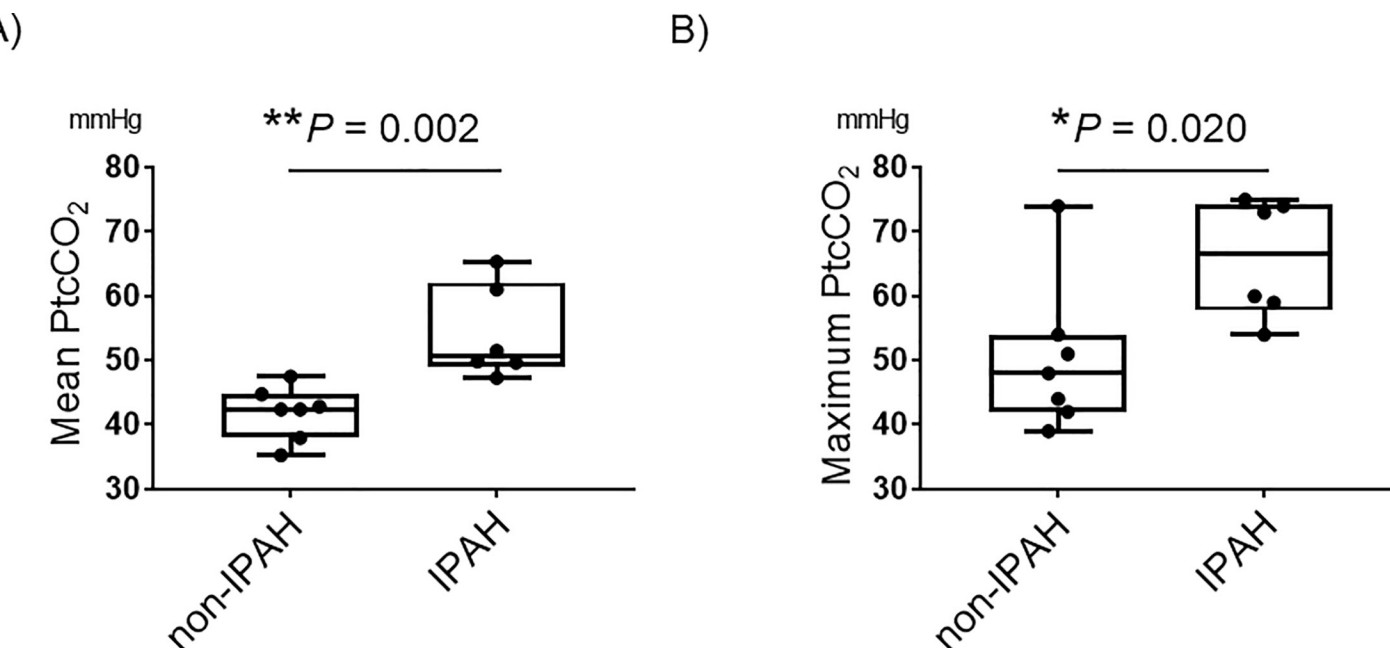

**Fig 1. Mean PtCO₂ in non-IPAH and IPAH patients.** Box plots indicating mean (A) or maximum (B) values of PtcCO₂ among IPAH (n = 6) or non-IPAH (n = 7) patients. Whiskers indicate the highest and lowest values. *P*-values were calculated using Mann-Whitney's U-test. *: P < 0.05, **: P < 0.01. IPAH: idiopathic pulmonary arterial hypertension.

due to systemic sclerosis, were only found in the patients without nocturnal hypercapnia, suggesting that these factors did not largely affect nocturnal hypercapnia in this study. Although we could not investigate muscle mass or muscle power directly, these pulmonary function data

**Table 2. Pulmonary function tests and blood gas indices.**

|  | Nocturnal hypercapnia (-) | Nocturnal hypercapnia (+) | *P*-value |
|---|---|---|---|
| Number | 7 | 6 | |
| Capnometer measurements | | | |
| 3%ODI (per hour) | 0.35 [0.00, 2.95] | 1.38 [0.12, 12.02] | 0.198 |
| Max PtcCO₂ (mmHg) | 48.00 [39.00, 54.00] | 73.50 [59.00, 75.00] | 0.003 |
| Mean PtcCO₂ (mmHg) | 42.41 [35.27, 47.32] | 50.70 [47.52, 65.36] | 0.003 |
| Pulmonary Function tests | | | |
| %FVC | 85.05 [56.00, 96.30] | 75.20 [38.10, 88.90] | 0.173 |
| %FEV1 | 79.80 [59.80, 87.20] | 65.20 [43.00, 87.10] | 0.109 |
| %DLco | 63.73 [31.57, 90.48] | 57.38 [32.74, 80.26] | 0.715 |
| %TLC | 94.10 [88.40, 112.10] | 84.35 [56.40, 92.20] | 0.018 |
| %PEF | 95.30 [78.80, 111.90] | 78.05 [70.20, 88.20] | 0.015 |
| Daytime arterial blood gas | | | |
| PaO₂ (mmHg) | 87.60 [41.00, 121.20] | 75.65 [56.50, 117.80] | 0.775 |
| PaCO₂ (mmHg) | 35.60 [28.70, 41.60] | 38.60 [32.80, 44.30] | 0.224 |
| HCO₃⁻ (mEq/l) | 22.50 [19.70, 27.10] | 25.35 [21.50, 27.00] | 0.199 |

Data are presented as median [range]. Mann-Whitney's U-test was used for statistical analyses.

FVC: forced vital capacity, FEV1: , DLco: , TLC: total lung capacity, PEF: peak expiratory flow, ODI: Oxygen desaturation index, PtcCO₂: transcutaneous carbon dioxide partial pressure.

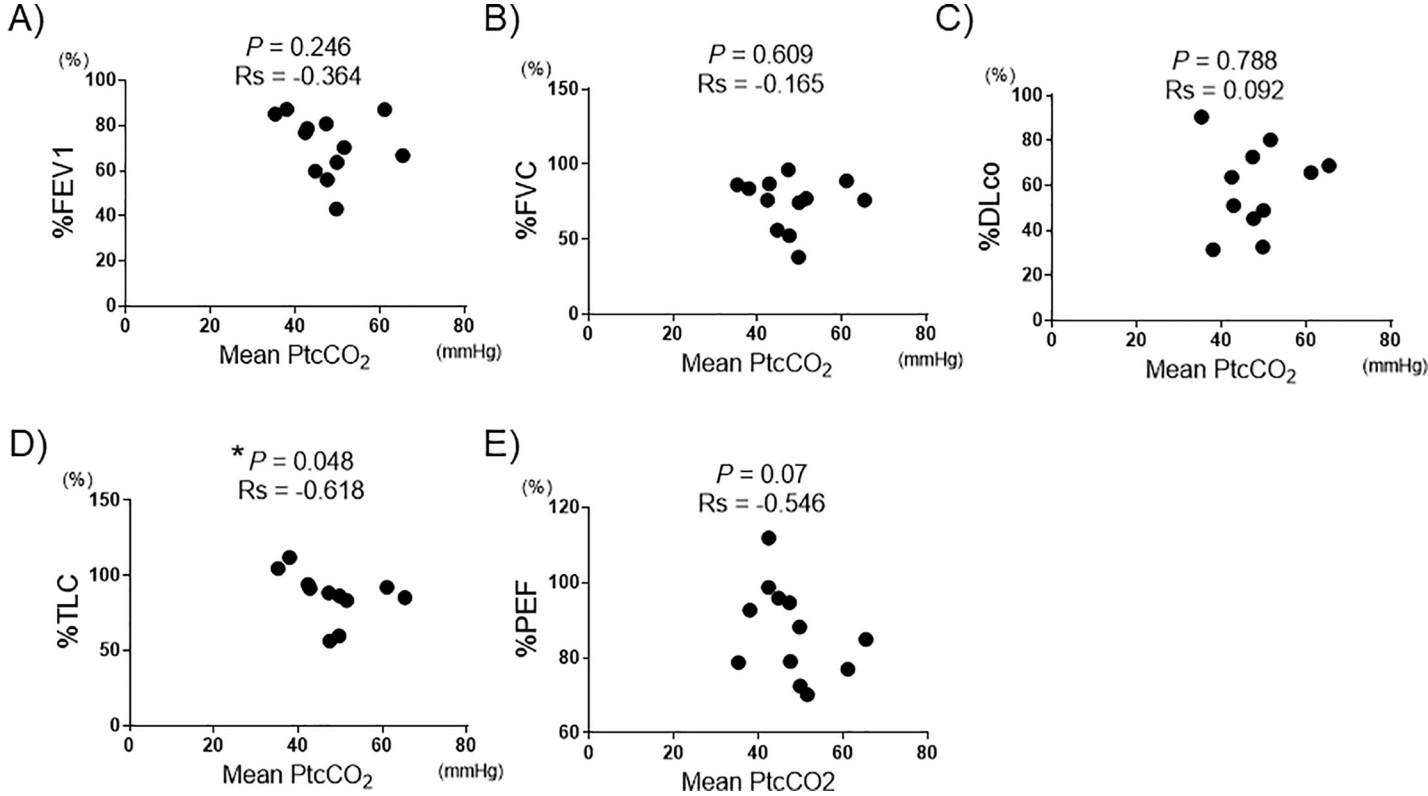

**Fig 2. Associations between PtCO$_2$ and pulmonary function test indices.** Scatter plots showing the correlations between mean PtcO$_2$ and %FEV1 (A), %FVC (B), % DLco (C), %TLC (D) or %PEF (E). Each dot represents one patient. Spearman's rank correlation test was used for statistical analysis. For one patient no pulmonary function data were available, and for two patients DLco and TLC data were not available. *: P < 0.05. FEV1: forced expiratory volume in one second, FVC: forced vital capacity, DLco: diffusing capacity for carbon monoxide, TLC: total lung capacity, PEF: peak expiratory flow.

imply that the lowered respiratory muscle power associated with a low BMI were candidate mechanisms for nocturnal hypercapnia. Recently, decreased muscle mass due to sarcopenia was shown to be frequent among lung transplantation candidates and constitute a risk factor for a poor prognosis [28]. In addition, inspiratory muscle power is substantially reduced during sleep, especially during rapid eye movement sleep [29]. Therefore, the reduction in minute

**Table 3. Right heart catheter measurements in the patients with and without nocturnal hypercapnia.**

|  | Nocturnal hypercapnia (-) | Nocturnal hypercapnia (+) | P-value |
|---|---|---|---|
| Number | 7 | 6 |  |
| Right heart catheter measurements |  |  |  |
| mean PAP (mmHg) | 54.00 [46.00, 82.00] | 57.00 [35.00, 92.00] | 0.774 |
| systolic PAP (mmHg) | 88.00 [62.00, 101.00] | 95.00 [52.00, 142.00] | 0.568 |
| diastolic PAP (mmHg) | 30.00 [15.00, 66.00] | 29.00 [23.00, 70.00] | 0.886 |
| mean PCWP (mmHg) | 9.00 [6.00, 25.00] | 11.00 [10.00, 13.00] | 0.126 |
| PVR (mmHg/L/min) | 10.32 [4.66, 23.27] | 6.31 [4.07, 24.05] | 0.317 |
| Cardiac Index (L/min/m$^2$) | 2.66 [1.98, 3.24] | 4.26 [2.66, 6.97] | 0.012 |
| Cardiac Output (L/min) | 3.97 [2.51, 5.25] | 6.54 [3.41, 8.43] | 0.015 |

Data are presented as median [range]. Mann-Whitney's U-test was used for statistical analyses.

PAP: pulmonary arterial pressure, PCWP: pulmonary capillary wedge pressure, PVR: pulmonary vascular resistance.

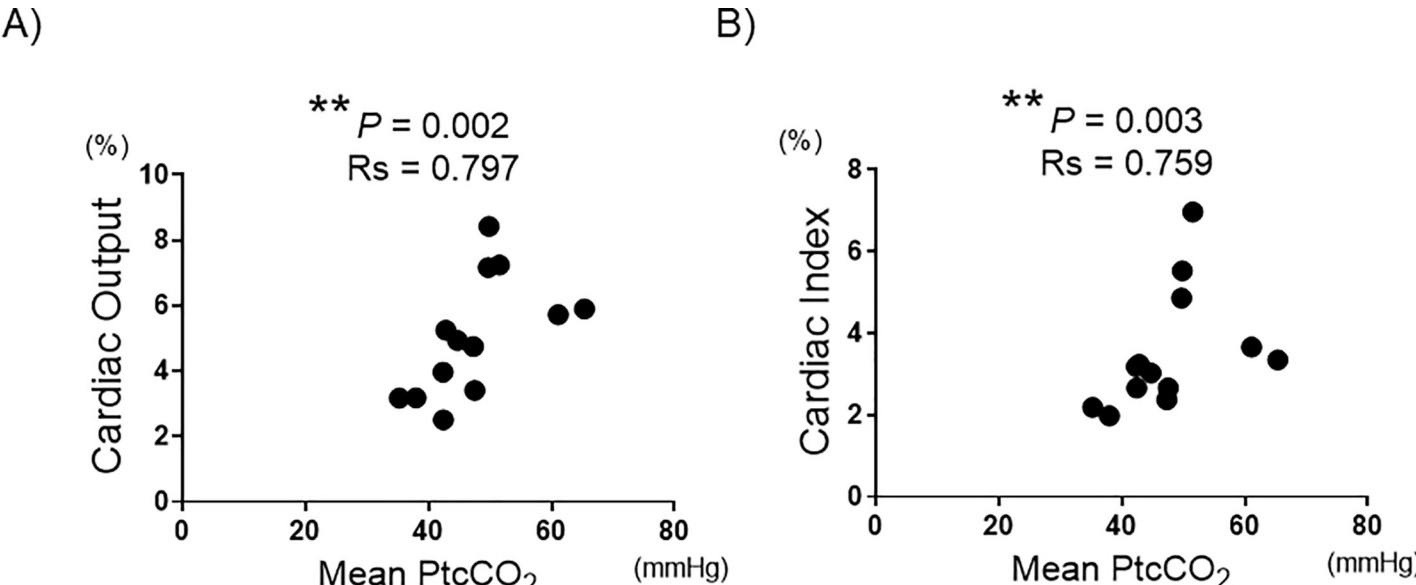

**Fig 3. Associations between $PtcCO_2$ and cardiac output or index.** Scatter plots showing the correlations between mean $PtcCO_2$ and cardiac output (A) or cardiac index (B). Each dot represents one patient. Spearman's rank correlation test was used for statistical analysis. **: P < 0.01.

ventilation which results in sleep-related hypoventilation might represent an early stage of chronic hypoventilation syndrome [11]. Indeed, the low values for %TLC in the study patients were associated with an elevation in mean $PtcCO_2$, which suggests relationships between impairment of pulmonary function due to the respiratory muscle dysfunction in IPAH patients and nocturnal hypercapnia. In addition to the impairment in pulmonary function, previous studies suggested that in PAH patients the cerebral microvascular blood flow response against $PaCO_2$ elevation is suppressed, thereby ameliorating the stimuli to breathe [30]. The integration of these factors may contribute to the high prevalence of nocturnal hypercapnia among IPAH patients. On the other hand, in the present study, there were no significant differences in BMI between patients with and without nocturnal hypercapnia, or between IPAH and non-IPAH patients. These results suggest that other factors had contributed to the impairment of muscle power especially in patient with nocturnal hypercapnia or IPAH patients. Therefore, further study is required to investigate the underlying pathophysiology of nocturnal hypercapnia in PAH patients.

In contrast to the high prevalence of nocturnal hypercapnia, daytime hypercapnia was not observed in any study patient, indicating that alveolar hypoventilation among PAH patients is largely limited during sleep. Recently, it was reported that noninvasive ventilation (NIV) can improve nocturnal hypercapnia in patients with nocturnal hypercapnia without daytime hypercapnia [31]. In addition, while alveolar hypoventilation may accompany low values for partial pressure of oxygen, the elevation of 3%ODI was rarely observed among the study patients, presumably because of treatment with oxygen supplementation. These results suggest that the assessment of either the daytime $PaCO_2$ level or nocturnal $SpO_2$ level is not sufficient to predict the complication of nocturnal hypercapnia in PAH patients, which highlights the significance of $PtcCO_2$ monitoring at night.

Hypoxia is regarded as a significant factor in the progression of PAH and careful monitoring is generally performed [2]. Indeed, in the present study oxygen supplementation was introduced to all study participants and oxygen desaturation was rarely observed. On the other hand, the clinical significance of nocturnal hypercapnia/hypocapnia in PAH patients has been

scarcely investigated. Physiologically, hypercapnia contributes to the escalation of PAP through several mechanisms. First, elevation of $PaCO_2$ directly induces the constriction of the pulmonary vasculature, thereby increases PVR [8]. Second, ventilatory acidosis resulting from rapid elevation of $PaCO_2$ also increases PVR [9]. Third, elevated $PaCO_2$ increases CO, which causes the escalation of PAP [10]. Even though nocturnal hypercapnia is reversible, prolonged and repeated exposure of the above factors eventually enhances stiffness of pulmonary vessels and results in an increase in PVR. Indeed, obesity hypoventilation syndrome is often complicated with PH [13], and NIV treatment during sleep can lower daytime PAP as well as PVR in these patients [17], suggesting that recurrent exposure to nocturnal hypercapnia is a treatable causality of daytime PH. Therefore, we anticipate that nocturnal hypercapnia among PAH patients may constitute a significant pathogenic role in the development of PAP escalation, and its normalization is a possible strategy for treatment of PAH.

In the present study, we found a significant positive correlation between mean $PtcCO_2$ levels at night and CO as well as CI. The increase in CO is a fundamental factor in the elevation of PAP; therefore it is suggested that nocturnal hypercapnia is associated with an increase in PAP. On the other hand, a correlation between mean $PtcCO_2$ at night and mean PAP or PVR was not apparent. We propose two possible interpretations of these data. One is that nocturnal hypercapnia increases CO without affecting PAP or PVR. The other interpretation is that nocturnal hypercapnia induces an increase in CO and PVR, but that in these patients the PVR elevation was masked by medication. We found that the usage of epoprostenol infusions was relatively frequent and also that the dose was higher in the study patients with nocturnal hypercapnia, which could have lowered PVR. A previous study showed that CO was well correlated lineally with $CO_2$ production (40). Therefore, patients with high CO would have higher $CO_2$ production than patients with relative low CO. In addition to low lung function (Table 2), the difference in $CO_2$ production related to CO would partly contribute to a significant elevation in mean and max $PtcCO_2$ in the patients with high CO: the patients with nocturnal hypercapnia (Tables 2 and 3). Because the current study could not dissect the contribution of nocturnal hypercapnia to the hemodynamic status in PAH patients without the confounding effect of medications, a future study can be expected to investigate whether nocturnal hypercapnia affects PAP or PVR. In addition, dynamic changes of PAP in parallel with the levels of $PtcCO_2$ were reported [13, 16]. Nocturnal hypercapnia was specifically observed during sleep, therefore a future study focusing on hemodynamics during sleep is expected.

Although the data sample was small, it is notable that as many as 46.2% (6/13) of PAH patients overall and 83.3% (5/6) of IPAH patients had nocturnal hypercapnia. We suppose that this high rate has been overlooked because these patients rarely manifest daytime hypercapnia. In general, the condition of PAH patients waiting for lung transplantation is refractory against full treatment with currently available medications. Considering that nocturnal hypercapnia is treatable with NIV [17, 31], we would consider that NIV during sleep could be a novel option to treat these patients. Further investigation of the use of NIV in such patients is needed.

This study has several limitations. First, due to its retrospective nature, disease severity, treatment procedures and comorbid diseases were not controlled for. In addition, because of the small number of study patients, we could not perform multivariate analysis to adjust for these variables. Second, the number of study patients was scant; thus the power to detect differences between groups may not be sufficient. Third, a detailed evaluation of sleep status (e.g. polysomnography) was not performed; therefore, precise events occurring in PAH patients during sleep were not analyzed. Especially, further investigations are required to reveal the pathophysiology affecting IPAH patients that contributed to the high frequency of nocturnal hypercapnia in this group.

In spite of these limitations, we for the first time reported that nocturnal hypercapnia was observed in nearly half of PAH patients and in the majority of IPAH patients who were considered for lung transplantation. Also revealed were possible correlations between nocturnal hypercapnia and impaired pulmonary function or altered hemodynamic status in the daytime. Further studies are needed to investigate hemodynamics during sleep and to dissect the direct effect of nocturnal hypercapnia on PAP or PVR and to evaluate whether the normalization of nocturnal hypercapnia has therapeutic significance for PAH patients.

## Supporting information

**S1 Fig. Representative TOSCA results for patients with nocturnal hypercapnia.** (A) TOSCA summary sheet for a 14-year-old female patient with IPAH and (B) 35-year-old female patient with IPAH. The intervals between the bold lines were analyzed. IPAH: idiopathic pulmonary arterial hypertension.
(TIF)

**S1 Dataset. The dataset used for the analyses in this study.**
(XLSX)

## Acknowledgments

The authors thank Y. Kohdono, S. Tamura and T. Toki (Department of Respiratory Care and Sleep Control Medicine, Graduate School of Medicine, Kyoto University) for their secretarial work to this research. The authors also thank NAI.Inc for editing the English text of this manuscript.

## Author Contributions

**Conceptualization:** Yoshinari Nakatsuka, Toyofumi Chen-Yoshikawa, Hideyuki Kinoshita, Hiroyasu Kubo, Kazuo Chin.

**Data curation:** Yoshinari Nakatsuka, Toyofumi Chen-Yoshikawa, Hideyuki Kinoshita, Akihiro Aoyama, Hiroyasu Kubo, Kimihiko Murase, Satoshi Hamada, Hirofumi Takeyama, Takuma Minami, Naomi Takahashi, Kiminobu Tanizawa, Tomohiro Handa, Toyohiro Hirai, Kazuo Chin.

**Formal analysis:** Yoshinari Nakatsuka, Toyofumi Chen-Yoshikawa, Hideyuki Kinoshita, Kazuo Chin.

**Funding acquisition:** Kazuo Chin.

**Investigation:** Yoshinari Nakatsuka, Kazuo Chin.

**Methodology:** Yoshinari Nakatsuka, Kazuo Chin.

**Project administration:** Kazuo Chin.

**Resources:** Kazuo Chin.

**Supervision:** Toyofumi Chen-Yoshikawa, Kazuo Chin.

**Writing – original draft:** Yoshinari Nakatsuka, Kazuo Chin.

**Writing – review & editing:** Toyofumi Chen-Yoshikawa, Hideyuki Kinoshita, Akihiro Aoyama, Kimihiko Murase, Satoshi Hamada, Hirofumi Takeyama, Takuma Minami, Naomi Takahashi, Toyohiro Hirai, Hiroshi Date, Kazuo Chin.

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
