## [Decision Letter · Decision Letter 0]

8 Oct 2019

PONE-D-19-24091

Nocturnal hypercapnia with daytime normocapnia in patients with advanced pulmonary arterial hypertension awaiting lung transplantation

PLOS ONE

Dear Dr. Chin,

Thank you for submitting your manuscript to PLOS ONE. After careful consideration, we feel that it has merit but does not fully meet PLOS ONE’s publication criteria as it currently stands. Therefore, we invite you to submit a revised version of the manuscript that addresses the points raised during the review process.

1, The manuscript needs to be edited by an English professional.

2, Address the questions raised by the reviewer, especially the size of the samples.

We would appreciate receiving your revised manuscript by six months. To enhance the reproducibility of your results, we recommend that if applicable you deposit your laboratory protocols in protocols.io, where a protocol can be assigned its own identifier (DOI) such that it can be cited independently in the future. For instructions see: http://journals.plos.org/plosone/s/submission-guidelines#loc-laboratory-protocols

We look forward to receiving your revised manuscript.

Kind regards,

Yunchao Su, MD, Ph.D.

Academic Editor

PLOS ONE

Journal Requirements:

2. In ethics statement in the manuscript and in the online submission form, please provide additional information about the patient records/samples used in your retrospective study. Specifically, please ensure that you have discussed whether all data/samples were fully anonymized before you accessed them and/or whether the IRB or ethics committee waived the requirement for informed consent. If patients provided informed written consent to have data/samples from their medical records used in research, please include this information.

'I have read the journal's policy and the authors of this manuscript have the following

competing interests: Yoshinari Nakatsuka and Naomi Takahashi reports grants from

Philips-Respironics, grants from ResMed Japan, grants from Fukuda Denshi, grants from Fukuda Lifetec Keiji. Kimihiko Murase and Hirofumi Takeyama reports grants from

Philips-Respironics, grants from Teijin Pharma, grants from Fukuda Denshi, grants

from Fukuda Lifetec Keiji. Kazuo Chin reports grants and personal fees from Philips-

Respironics, grants and personal fees from Teijin Pharma, grants and personal fees

from Fukuda Denshi, grants and personal fees from Fukuda Lifetec Keiji, grants from

KYORIN Pharmaceutical Co., Ltd, grants from Nippon Boehringer Ingelheim Co., Ltd,

grants and personal fees from GlaxoSmithKline, personal fees from MSD, personal

fees from Resmed, personal fees from Astellas Pharma, personal fees from Eisai Co.,

Ltd. Hideyuki Kinoshita reports personal fees from Actelion Pharmaceuticals Japan

Ltd., Nippon Shinyaku Co., Ltd, Bayer Yakuhin, Ltd. and research grant from Bayer

Yakuhin, Ltd. Toyofumi Chen-Yoshikawa, Akihiro Aoyama, Hiroyasu Kubo, Satoshi

Hamada, Takuma Minami, Kiminobu Tanizawa, Tomohiro Handa, Toyohiro Hirai and

Hiroshi Date declare no potential conflict of interests.

The Department of Respiratory Care and Sleep Control Medicine is funded by

endowments from Philips-Respironics, ResMed, Fukuda Denshi and Fukuda Lifetec-

Keiji to Kyoto University.'

Please confirm that this does not alter your adherence to all PLOS ONE policies on sharing data and materials, by including the following statement: "This does not alter our adherence to  PLOS ONE policies on sharing data and materials.” (as detailed online in our guide for authors http://journals.plos.org/plosone/s/competing-interests).

If there are restrictions on sharing of data and/or materials, please state these. Please note that we cannot proceed with consideration of your article until this information has been declared.

Additional Editor Comments (if provided):

Reviewers' comments:

Reviewer's Responses to Questions

**Comments to the Author**

1. Is the manuscript technically sound, and do the data support the conclusions?

Reviewer #1: Partly

2. Has the statistical analysis been performed appropriately and rigorously? 

Reviewer #1: Yes

3. Have the authors made all data underlying the findings in their manuscript fully available?

Reviewer #1: Yes

4. Is the manuscript presented in an intelligible fashion and written in standard English?

Reviewer #1: Yes

5. Review Comments to the Author

Reviewer #1: Answer 1: See below

Answer 2: Yes, but should be justified (see comment below)

Answer 3: Yes, the authors indicate that the individual data sets are available only upon request; they were not available in the supplemental information or included in the original submission.

Answer 4: Yes, but there are scattered, small grammatical and document formatting errors.

-----

In the submission entitled, "Nocturnal hypercapnia with daytime normocapnia in patients with advanced pulmonary arterial hypertension (PAH) awaiting lung transplantation," the authors describe a small cohort of patients with PAH and the incidence of nocturnal hypercapnia in this population, speculating as to the significance of this finding, as well as if it offers a possible additional therapeutic intervention in these patients.

General Comments

1. There are scattered small, but notable grammatical errors throughout the manuscript. If revised, it would benefit from ensuring that these are corrected.

Comments on the Manuscript

1. In general, the authors overstate their conclusions re: this population (e.g. "it is notable that as many as 46.2% of PAH patients and 83.3% of IPAH patients were complicated by nocturnal hypercapnia") given that the sample size / retrospective review involved such a few number of patients. When the numbers themselves are examined, this study only looked at 13 patients and each group (nocturnal hypercapnia (-) versus nocturnal hypercapnia (+) only contained 7 and 6 patients, respectively. It is very difficult to draw the firm conclusions these authors have based on these numbers alone as it is very doubtful that these small #s would allow this review to be sufficiently powered to detect the indicated differences

2. In the "Methods" section, the authors indicate the specific statistical tests used, but should be more specific (e.g. "this test was used as the results demonstrate a non-normal distribution" or other justification)

3. The authors comment that "no study has investigated the impact of nocturnal hypercapnia on refractory PAH patients so far," yet this study does not meet this objective; it just describes the incidence of nocturnal hypercapnia, but does not show that it actually leads to worse PAH (at least per indicated right heart catheterization numbers provided) or leads PAH that is necessarily more refractory to treatment. It also does not appear to necessarily change outcomes post-transplantation (although again, the number of patients is small)

4. In the discussion, the authors postulate that "lung and chest wall elastic recoil are not changed in patients with PAH" which is true, but does not take into account that each of these individual patients may have had another factor(s) that could have modified these variables (for example, prior chest surgery) and contributed to decline in TLC (not addressed).

5. The authors also postulate that a low BMI "implied the existence of lower respiratory muscle mass" - but does not give a source / animal model / other to support this conclusion. In addition, they imply that this occurs more in the IPAH patients they studied, but do not provide this data (e.g. comparison of BMI between IPAH and non-IPAH). Other data that may support this claim would be NIF / NEF as respiratory muscle weakness (if present) should also be detectable while awake

Comments on the Figures

1. Figures should include significance markers (*) per convention in addition to the p-values.

2. In Figure 1 for the box-and-whisker plots, please include the individual data points within the plots and the number of patients per group. In addition, the Y-axis titles on both graphs are labeled "mean," however, the text of the figures indicates one graph is the mean PtcCO2, while the other is the maximum PtcCO2.

6. PLOS authors have the option to publish the peer review history of their article (what does this mean?). If published, this will include your full peer review and any attached files.

Reviewer #1: No

---

## [Author Response · Author response to Decision Letter 0]

22 Dec 2019

We thank the reviewer for his/her constructive comments. Below please find our point-by-point responses to all comments. We believe that we have examined each of the points raised and responded thoroughly. We would thank the reviewer for careful examination of our manuscript and thoughtful suggestions, which have definitely helped us to make this manuscript a better paper.

Detailed replies to specific comments are shown below.

General Comments

1. There are scattered small, but notable grammatical errors throughout the manuscript. If revised, it would benefit from ensuring that these are corrected.

Author reply:

Thank you for the thorough check of our grammatical errors. To minimize the possibility of further errors, we have consulted a company specialized in editing medical and scientific materials written by those whose first language is not English (NAI, Inc.). We mentioned appreciation of this review in the “Acknowledgments” section.

“The authors also thank NAI.Inc for editing the English text of this manuscript.”

Comments on the Manuscript

1. In general, the authors overstate their conclusions re: this population (e.g. "it is notable that as many as 46.2% of PAH patients and 83.3% of IPAH patients were complicated by nocturnal hypercapnia") given that the sample size / retrospective review involved such a few number of patients. When the numbers themselves are examined, this study only looked at 13 patients and each group (nocturnal hypercapnia (-) versus nocturnal hypercapnia (+) only contained 7 and 6 patients, respectively. It is very difficult to draw the firm conclusions these authors have based on these numbers alone as it is very doubtful that these small #s would allow this review to be sufficiently powered to detect the indicated differences

Author reply:

We appreciate the comment. To clarify the prevalence of nocturnal hypercapnia among PAH patients, we added the description of actual number of total patient number as well as the number of patients with nocturnal hypercapnia as follows;

(Abstract “Results” section, page: 2, line:13)

Of note, nocturnal hypercapnia was found for 5 out of 6 patients with idiopathic PAH (83.3%).

(Manuscript, “Results” section, page: 8, line: 8)

Notably, of the 6 patients with nocturnal hypercapnia, 5 had IPAH (83.3%).

We agree that there is a difficulty in reaching a firm conclusion with this small number of patients. Because the background of patients such as treatment procedures, age, sex and BMI varied, the adjustment of these factors through multivariate analysis would be desired, but the patient number did not allow us to do so. To clearly recognize this shortcoming, we newly added the following sentence as a limitation;

(Manuscript, “Discussion” section, page: 16, line: 16)

In addition, because of the small number of study patients, we could not perform multivariate analysis to adjust these variables.

On the other hand, we would like to emphasize that these patients were strictly selected from a large number of patients (246 patients) who were considered for lung transplantation in Kyoto University. As Kyoto University performs the largest number of lung transplantation surgery in Japan, and pre-operative evaluation of PtcCO2 is not a regular practice in institutions other than Kyoto University, we think this study included the largest number of patients currently possible Further validation of the results requires a prospective study ideally of a larger number of patients. In spite of the small number of patients, this study includes a number of clinically relevant results that were statistically significant and were supported by previous human experimental studies (e.g. the association between PtcCO2 values and cardiac output). Therefore, we believe that the data presented in this manuscript will motivate clinicians to conduct the future prospective studies to dissect the clinical impact of nocturnal hypercapnia among PAH patients.

2. In the "Methods" section, the authors indicate the specific statistical tests used, but should be more specific (e.g. "this test was used as the results demonstrate a non-normal distribution" or other justification)

Author reply:

It is obviously beneficial to clarify the reason why we used a specific statistical test, so we appreciate the reviewer’s suggestion of this modification. As the reviewer pointed out and we agreed, the number of study patients was small. For this reason we considered that it was not appropriate to determine whether each parameter followed normal distribution through statistical tests such as the Kolmogorov–Smirnov test and Shapiro-Wilk test. Therefore we decided to use non-parametric statistics for the analyses throughout the study. We added the following sentence to the “Methods” section.

(Manuscript, “Method” section, page: 7, line: 5)

“We considered that the number of study patients was too small to determine whether each parameter followed normal distribution or not, therefore we employed non-parametric statistics for analyses.”

3. The authors comment that “no study has investigated the impact of nocturnal hypercapnia on refractory PAH patients so far,” yet this study does not meet this objective; it just describes the incidence of nocturnal hypercapnia, but does not show that it actually leads to worse PAH (at least per indicated right heart catheterization numbers provided) or leads PAH that is necessarily more refractory to treatment. It also does not appear to necessarily change outcomes post-transplantation (although again, the number of patients is small)

Author reply:

We are grateful for this thoughtful comment. We think this criticism largely arises from the fact that this retrospective study does not include longitudinal data that would support a causal relationship between nocturnal hypercapnia and clinical outcomes. On this point, we would like to note that this study focused on the associations between PtcCO2 values and other clinical parameters, which at least suggested the relationship between these factors. We agree that the word “impact” could have been misleading, therefore, we modified the descriptions as follows;

(Manuscript, “Introduction” section, page: 5, line: 5)

“However, thus far no study has investigated the associations between nocturnal hypercapnia and clinical indices for refractory PAH patients so far.”

On the other hand, in “Discussion” section, we described the potential of nocturnal hypercapnia as a therapeutic target. This consideration was based on the results of this study as well as a previous physiological experimental study (Westcott RN, et al. J Clin Invest. 1951) and prospective study revealing the effect of NPPV on lowering the PAP (Held M, et al. Eur Respir J. 2014). Therefore we retained the description that was in the original version of this manuscript.

4. In the discussion, the authors postulate that "lung and chest wall elastic recoil are not changed in patients with PAH" which is true, but does not take into account that each of these individual patients may have had another factor(s) that could have modified these variables (for example, prior chest surgery) and contributed to decline in TLC (not addressed).

Author reply:

We appreciate the comment. In this study, two patients had undergone prior chest surgery. Both surgeries were repair surgery for congenital heart diseases. We agree that the surgical procedure could have reduced the mobility of the chest wall and thereby contributed to the limitation of TLC. We added these data to Table 1. Another potential factor for low TLC was the skin stiffness of a patient with systemic sclerosis, which could restrict chest movement. However, in this study, these patients did not have nocturnal hypercapnia. Therefore, we think that the contribution of these factors would not be large for the development of nocturnal hypercapnia.

Based on these considerations, we modified the description as follows:

(Manuscript, “Results” section, page: 8, line: 3)

“Two patients had received prior chest surgery, both of which were repair surgery for congenital heart diseases, neither of these patients exhibited nocturnal hypercapnia.”

(Manuscript, “Discussion” section, page: 12, line: 9)

“Other potential factors for the restriction of chest movement, such as previous thoracic surgery or skin stiffness due to systemic sclerosis, were only found in the patients without nocturnal hypercapnia, suggesting that these factors did not largely affect nocturnal hypercapnia in this study.”

5. The authors also postulate that a low BMI "implied the existence of lower respiratory muscle mass" - but does not give a source / animal model / other to support this conclusion. In addition, they imply that this occurs more in the IPAH patients they studied, but do not provide this data (e.g. comparison of BMI between IPAH and non-IPAH). Other data that may support this claim would be NIF / NEF as respiratory muscle weakness (if present) should also be detectable while awake

We appreciate this constructive criticism and thoughtful suggestion. The association between low BMI and the decline in respiratory muscle mass has been reported (Arora NS, et al. Am Rev Respir Dis. 1982), and concordantly, a proportional decline in respiratory pressure for the individuals with low BMI was shown (Sgariboldi D, et al. Respir Care. 2016). These studies clearly suggest that low BMI was associated with low respiratory muscle mass as well as impairment of respiratory muscle function. 

Unfortunately, in the present study the data for muscle mass, NIF or NEF were not available. Instead, we additionally analyzed the pulmonary function tests according to the reviewer’s suggestion and found that percent predicted peak expiratory flow (%PEF) was significantly lower for those with nocturnal hypercapnia. As %PEF is a useful indicator of expiratory flow pressure especially for those who do not have airway flow limitation, in the present study there was no patient who was complicated by an obstructive disorder. We think that this finding supports the decreased muscle strength in patients with nocturnal hypercapnia. We added the data to Table 2 and Figure 2, and modified the description in “Results” and “Discussion” sections as follows.

(Manuscript, “Results” section, page: 12, line: 1)

“In the present study, low values for %FVC, %TLC and %PEF were the most distinct features of nocturnal hypercapnia. Total lung capacity is determined by lung elastic recoil, chest wall elastic recoil and inspiratory muscle power, whereas expiratory muscle power and airway obstruction largely affects PEF [27]. Given that in this study restrictive and obstructive pulmonary diseases were denied by computed tomography and pulmonary function test, and lung and chest wall elastic recoils are not changed in patients with PAH in general, the decrease in %TLC and %PEF would be due to a decline in inspiratory and expiratory muscle power, respectively[27].”

Whereas obesity hypoventilation is one of the most frequent causalities for nocturnal hypercapnia, based on the considerations above, we think low BMI could also cause hypoventilation due to the impairment of muscle power. We added the following sentences to the “Discussion” section, and newly added references. 

(Manuscript, “Discussion” section, page: 12, line: 13)

“Although obesity hypoventilation syndrome is a major cause of nocturnal hypercapnia[11], the BMI in the majority of the current study patients was lower than normal (18.5 kg/m2). Previously it was revealed that a low BMI affects the respiratory muscle mass, which results in the impairment of muscle performance[25] and a proportional decline in respiratory pressure[26].”

(Manuscript, “Discussion” section, page: 13, line: 11)

“Although we could not investigate muscle mass or muscle power directly, these pulmonary function data imply that the lowered respiratory muscle power associated with a low BMI were candidate mechanisms for nocturnal hypercapnia.”

On the other hand, we did not find a statistically significant difference in BMI between IPHA patients and non-IPAH patients. We found that IPAH patients had a relatively higher frequency of %PEF < 90% (5 out of 6 in IPAH and 2 out of 6 in non-IPAH), which implies the higher risk of muscle force impairment among IPAH patients compared with non-IPAH patients who had equivalent BMIs, although this difference was not statistically significant (P = 0.242). In addition, we also found no difference in BMI between patients with and without nocturnal hypercapnia as shown in Table 1. As we described in the manuscript, nocturnal hypercapnia could arise from complex mechanisms including impairment of pulmonary function and insufficient response against hypoxia/hypercapnia. Therefore further investigation is required to clarify the underlying mechanisms for the high frequency of nocturnal hypercapnia among IPAH patients and factors causing the impairment of respiratory muscle power. Based on these considerations, we added the following to the “Discussion” section:

(Manuscript, “Discussion” section, page: 14, line: 2)

“On the other hand, in the present study, there were no significant differences in BMI between patients with and without nocturnal hypercapnia, or between IPAH and non-IPAH patients. These results suggest that other factors had contributed to the impairment of muscle power especially in patient with nocturnal hypercapnia or IPAH patients. Therefore, further study is required to investigate the underlying pathophysiology of nocturnal hypercapnia in PAH patients.”

(Manuscript, “Discussion” section, page: 16, line: 21)

“Especially, further investigations are required to reveal the pathophysiology affecting IPAH patients that contributed to the high frequency of nocturnal hypercapnia in this group.”

Comments on the Figures

1. Figures should include significance markers (*) per convention in addition to the p-values.

Author reply:

We appreciate the suggestion. According to the reviewer’s comment, we added the markers for all the figures.

2. In Figure 1 for the box-and-whisker plots, please include the individual data points within the plots and the number of patients per group. In addition, the Y-axis titles on both graphs are labeled "mean," however, the text of the figures indicates one graph is the mean PtcCO2, while the other is the maximum PtcCO2.

Author reply:

We thank for the reviewer to find out our error. We corrected the Y-axis titles of the right panel in Figure 1 as “maximum PtcCO2”. In addition, we modified the figure to show the individual data overlaying the box-and-whisker plots.

---

## [Editor Report · Decision Letter 1]

30 Dec 2019

Nocturnal hypercapnia with daytime normocapnia in patients with advanced pulmonary arterial hypertension awaiting lung transplantation

PONE-D-19-24091R1

Dear Dr. Chin,

We are pleased to inform you that your manuscript has been judged scientifically suitable for publication and will be formally accepted for publication once it complies with all outstanding technical requirements.

With kind regards,

Yunchao Su, MD, Ph.D.

Academic Editor

PLOS ONE
---

## [Editor Report · Acceptance letter]

7 Jan 2020

PONE-D-19-24091R1 

Nocturnal hypercapnia with daytime normocapnia in patients with advanced pulmonary arterial hypertension awaiting lung transplantation 

Dear Dr. Chin:

I am pleased to inform you that your manuscript has been deemed suitable for publication in PLOS ONE. Congratulations! Your manuscript is now with our production department. 

With kind regards,

on behalf of

Dr Yunchao Su 

Academic Editor

PLOS ONE